# Extracellular Citrate Treatment Induces HIF1α Degradation and Inhibits the Growth of Low-Glycolytic Hepatocellular Carcinoma under Hypoxia

**DOI:** 10.3390/cancers14143355

**Published:** 2022-07-10

**Authors:** Seon Yoo Kim, Dongwoo Kim, Jisu Kim, Hae Young Ko, Won Jin Kim, Youngjoo Park, Hye Won Lee, Dai Hoon Han, Kyung Sik Kim, Sunghyouk Park, Misu Lee, Mijin Yun

**Affiliations:** 1Department of Nuclear Medicine, Severance Hospital, Yonsei University College of Medicine, Seoul 03722, Korea; sykim1703@yuhs.ac (S.Y.K.); kdwoo@yuhs.ac (D.K.); wisdom7045@yuhs.ac (J.K.); hyko23@yuhs.ac (H.Y.K.); sarahyjpark@yonsei.ac.kr (Y.P.); 2Division of Life Sciences, College of Life Science and Bioengineering, Incheon National University, Incheon 22012, Korea; katips@inu.ac.kr; 3Department of Internal Medicine, Severance Hospital, Yonsei University College of Medicine, Seoul 03722, Korea; lorry-lee@yuhs.ac; 4Department of Surgery, Severance Hospital, Yonsei University College of Medicine, Seoul 03722, Korea; dhhan@yuhs.ac (D.H.H.); kskim88@yuhs.ac (K.S.K.); 5Department of Manufacturing Pharmacy, Natural Product Research Institute, College of Pharmacy, Seoul National University, Seoul 08826, Korea; psh@snu.ac.kr; 6Institute for New Drug Development, College of Life Science and Bioengineering, Incheon National University, Incheon 22012, Korea

**Keywords:** *SLC13A5*/NaCT, citrate, cancer metabolism, hepatocellular carcinoma, hypoxia

## Abstract

**Simple Summary:**

Patients with low-glycolytic hepatocellular carcinoma (HCC) show better clinical outcomes than those with hypoxic and high-glycolytic HCC. Low-glycolytic HCCs seem to utilize carbon sources other than glucose for metabolic fuel and tumor growth. However, by increasing tumor size, its outgrowth perfusion generates hypoxic foci inside the tumor and becomes more aggressive and resistant to therapy. In this study, we found that *SLC13A5*/NaCT is an important solute carrier (SLC) in low-glycolytic HCCs. To adapt to hypoxic conditions, low-glycolytic cancer cells have to switch metabolism from oxidative phosphorylation to hypoxia-induced glycolysis by the upregulation of HIF1α. However, extracellular citrate treatment in HCCs with high *SLC13A5*/NaCT expression had reduced glucose uptake due to HIF1α degradation, inducing the failure of metabolic adaptation to hypoxia, resulting in anti-cancer effects in in vitro and in vivo animal models.

**Abstract:**

HCC is well known for low glycolysis in the tumors, whereas hypoxia induces glycolytic phenotype and tumor progression. This study was conducted to evaluate the expression of SLCs in human HCCs and investigated whether extracellular nutrient administration related to SLCs in low-glycolytic HCC can prevent hypoxic tumor progression. SLCs expression was screened according to the level of glycolysis in HCCs. Then, whether extracellular nutrient treatment can affect hypoxic tumor progression, as well as the mechanisms, were evaluated in an in vitro cell line and an in vivo animal model. Low-glycolytic HCCs showed high *SLC13A5*/NaCT and *SLC16A1*/MCT1 but low *SLC2A1*/GLUT1 and *HIF1α*/HIF1α expression. Especially, high *SLC13A5* expression was significantly associated with good overall survival in the Cancer Genome Atlas (TCGA) database. In HepG2 cells with the highest NaCT expression, extracellular citrate treatment upon hypoxia induced HIF1α degradation, which led to reduced glycolysis and cellular proliferation. Finally, in HepG2-animal models, the citrate-treated group showed smaller tumor with less hypoxic areas than the vehicle-treated group. In patients with HCC, *SLC13A5*/NaCT is an important SLC, which is associated with low glycolysis and good prognosis. Extracellular citrate treatment induced the failure of metabolic adaptation to hypoxia and tumor growth inhibition, which can be a potential therapeutic strategy in HCCs.

## 1. Introduction

Hepatocellular carcinoma (HCC) is the sixth most common diagnosed cancer and the third leading cause of cancer-related mortality worldwide [1]. HCC undergoes a characteristic hemodynamic transition from a predominantly portal to a solely arterial blood supply during tumor progression. As a result, HCC growth outpaces perfusion despite increased arterial flow, which eventually leads to hypoxia and progression to an aggressive tumor [2]. Hypoxia in HCCs causes differences in metabolic phenotypes on ^18^F-fluorodeoxyglucose (FDG) positron emission tomography/computed tomography (PET/CT), a functional imaging modality for tumor glycolysis [3]. While hypoxic tumor regions of HCC are associated with increased glycolysis on ^18^F-FDG PET/CT (high-glycolytic HCC), tumors without hypoxia show low-level ^18^F-FDG uptake (low-glycolytic HCC) [2]. More importantly, an increased glycolytic phenotype on PET/CT is associated with large tumor size, high histological grade, and poor clinical outcomes [4,5,6]. Therefore, the inhibition of glycolytic metabolism can effectively prevent tumor progression in HCCs.

Cell requires sufficient nutrient supply, including carbon and nitrogen sources, to support growth and survival [7]. The solute carrier (SLC) group of membrane transport proteins participates as a metabolic gateway in cellular nutrient uptake [8]. Particularly, the use of glucose is important for the generation of adenosine triphosphate (ATP) and building blocks for cellular proliferation [9]. Thus, cancer cells induce changes in metabolic programs from oxidative phosphorylation to glycolysis through increased expression of the glucose transporter [10]. However, low-glycolytic tumors seem to utilize carbon sources other than glucose for bioenergetics and tumor proliferation [9,11]. Furthermore, acetate uptake through the increased expression of monocarboxylate transporter 1 (MCT1) is an important factor in generating Acetyl-CoA for energy production (via the tricarboxylic acid (TCA) cycle) and biomass in low-glycolytic HCCs [12,13,14]. Thus, ^11^C-acetate, a radioactive version of acetate, has been proposed as a complementary radiotracer for detecting low-glycolytic HCCs that cannot be identified by ^18^F-FDG [15]. However, apart from acetate, there is limited information on other carbon sources for tumor growth in low-glycolytic HCCs. Furthermore, low-glycolytic HCCs depend on oxidative phosphorylation using alternative carbon sources other than glucose. Citrate is a well-known tricarboxylic acid (TCA) cycle intermediate and an important carbon source for the biosynthesis of fatty acid and cholesterol synthesis [16,17]. The pool of cellular citrate was previously believed to be derived mainly from the reverse TCA cycle through glucose-derived pyruvate and reductive carboxylation of glutamine [18]. However, several lines of evidence suggest that the uptake and utilization of citrate from the extracellular space may play a role in cancer biology [19,20,21]. An elevated cytosolic citrate concentration inhibits the growth of several types of human cancer cells [22,23,24].

To investigate the utilization of carbon sources in low-glycolytic HCCs, we compared the expression of SLCs in human HCCs with low versus high glucose uptake. Then, in HCCs with low glycolysis, the most expressed SLC and its metabolic substrate was selected to investigate its effects on glucose metabolism and tumor growth in vitro and in vivo.

## 2. Materials and Methods

### 2.1. Human HCC Samples

The use of tissue samples for this study was approved by the Institutional Review Board at Yonsei University Health System Severance Hospital (Seoul, Korea, Yonsei IRB number 4-2015-0904). This study was conducted according to the current guidelines for ethical research. All patients provided oral and written consent after receiving detailed information on the study and agreed to data collection. Our study included patients with CC treated by curative surgical resection who had undergone ^18^F-FDG scans for preoperative staging at our institution. Gene expression profiles and protein expression for SLCs were compared between high and low ^18^F-FDG tumors. HCC with ^18^F-FDG uptake higher than that in the background liver was classified as a high-glycolytic tumor.

### 2.2. Cell Culture

HepG2 cells were cultured in RPMI-1640 (Hyclone, Logan, UT, USA), whereas SK-Hep1 and Huh7 were cultured in Dulbecco’s Modified Eagle Medium (Hyclone, Logan, UT, USA). All media were supplemented with 10% fetal bovine serum (Thermo Fisher, Waltham, MA, USA) and 1% penicillin/streptomycin (Thermo Fisher, Waltham, MA, USA). All cell lines were purchased from the Korean Cell Line Bank (http://cellbank.snu.ac.kr, accessed on 14 July 2020). In addition, a predesigned siRNA targeting hypoxia-inducible factor 1α (HIF1α) and mouse double minute 2 homolog (MDM2) was purchased from Bioneer (Daejeon, Korea). The cells were transfected using Lipofectamine RNAi MAX (Invitrogen, cat #13778-075, Waltham, MA, USA) following the manufacturer’s instructions. To induced hypoxic conditions, we used a hypoxia incubator (1% O_2_) or 100 μM CoCl_2_, depending on the experiment, since Western blotting (WB) showed that both conditions resulted in a similar expression of HIF1α (Appendix A).

### 2.3. In Vitro ^14^C-Deoxyglucose (DG) and ^14^C-Citrate Uptake Assay

HepG2 cells were incubated with sodium citrate (1 or 5 mM) and CoCl_2_ (100 μM, Sigma-Aldrich, St. Louis, MO, USA). After incubating for 24 h, cells were treated for 1 h with 0.5 μCi ^14^C-DG (Perkin Elmer, Waltham, MA, USA) in DMEM medium (Gibco, cat # 11966-025, Waltham, MA, USA) with 5 mM glucose, or 0.1 μCi ^14^C-citrate (PerkinElmer) in a reaction buffer containing 25 mM HEPES (pH 7.5), 140 mM sodium chloride, 5.4 mM potassium chloride, 0.8 mM magnesium sulfate, 1.8 mM calcium chloride, 25 mM Tris-HCl, and 5 mM D-glucose for 2 h. After washing with PBS, the cells were lysed using 0.1 N NaOH lysis buffer (Sigma-Aldrich) for 2 h. The cell lysates were collected in 6 mL pony vials (Perkin Elmer) containing 5 mL Ultima Gold liquid scintillation cocktail (Perkin Elmer) and measured using a liquid scintillation counter (Perkin Elmer). The measured count per minute was normalized to the protein concentration determined using a bicinchoninic acid protein assay kit (Thermo Fisher, Waltham, MA, USA).

### 2.4. Cell Viability and ATP Measurement

HepG2 cells were treated with sodium citrate (1 or 5 mM) under hypoxia. After 24 h, the cell viability of HepG2 cells was estimated using a cell counting kit (CCK-8). The absorbance was measured at a wavelength of 450 nm. ATP levels were determined using an ATP assay kit (Abcam, Cambridge, UK) according to the manufacturer’s instructions. Cell viability (absorbance: 562 nm) and ATP (absorbance: 570 nm) were measured using a microplate reader (Molecular Devices, Sunnyvale, CA, USA). All experiments were performed at least three times with three wells in each condition.

### 2.5. Western Blotting

Total protein was extracted using sodium dodecyl sulfate (SDS) lysis buffer (1% SDS, 60 mM Tris-HCl) with protease inhibitor (Roche) and phosphatase inhibitor (PhosSTOP, Roche, Basel, Switzerland). ImageQuant LAS 4000 mini (GE, Chicago, IL, USA) was used for the digital visualization of the chemiluminescent Western blots. These experiments were replicated at least three times with similar results. ImageJ (National Institutes of Health, Bethesda, Maryland, USA) software (ver 1.4.3) was used for band quantification. The primary antibodies in the present study for WB are listed in Appendix A.

### 2.6. Flow Cytometry

HepG2 cells (1 × 10^6^ cells/mL) were prepared following the manufacturer’s instructions to measure the apoptotic rate using the FITC Annexin V apoptosis detection kit I (BD science) and mitochondrial reactive oxygen species (ROS) using the MitoSOX (Invitrogen, Waltham, MA, USA). Apoptotic rate and ROS were quantitatively analyzed using the BD FACS LSR II SORP system (BD science, Franklyn-lake, NJ, USA). Details of the gating strategy for analysis are shown in Appendix A.

### 2.7. In Vivo Experiment Using HepG2-Xenograft Animal Model

All animal experiments were approved by The Institutional Animal Care and Use Committee of the Yonsei University Health System (approval number: 2020-0242). Four-to-five-week-old female BALB/c nude mice were purchased from Orient Bio (Gyeonggi-do, Korea). HepG2 cells (1 × 10^7^ cells 100 μL PBS) mixed with 100 μL matrigel were subcutaneously injected into the right leg of each mouse. Tumor size was calculated as 1/2 × A × B^2^ (where A is the long axis, and B is the short axis). When the tumor size reached 300 mm^3^, the mice were allowed to drink water supplemented with the vehicle (PBS) or citrate (4 g/kg daily) for 28 days.

### 2.8. RNA Expression

The Cancer Genome Atlas (TCGA) data were obtained from the OncoLnc TCGA data portal (www.oncolnc.org, accessed on 14 July 2020). A set of 360 HCC samples with high and low gene expression groups (30–70 percentile) was used for analysis. GraphPad Prism 9 (San Diego, CA, USA) was used for mapping. Total RNAs were extracted from human HCC tissues and cell lines using a RNeasy Mini Kit (QIAGEN, Munich, Germany) for RNA expression profiling. RNA sequencing and data analysis were assessed by EBIOGEN Inc. (Seoul, Korea).

### 2.9. Statistical Analysis

Statistical analyses were performed using GraphPad Prism software (GraphPad Software, Inc. (California, CA, USA)). Results were expressed as means ± SEM (range). The *p*-values between groups were calculated using one-way analysis of variance (ANOVA) with Tukey’s multiple comparison test in multiple groups and unpaired *t*-tests in two groups. A log-rank test was conducted to compare overall survival (OS). Pearson’s correlation coefficient was used to calculate the correlation between the two factors. A *p*-value of less than 0.05 was considered significant.

## 3. Results

### 3.1. Correlation of SLC-Related Genes and Proteins in HCC Patients with High- and Low-Glycolysis

We analyzed the expression of SLC genes between high- and low-glycolytic tumors. SLCs are major nutrient transporters exhibiting various expression levels in cancers with different metabolic phenotypes. Our gene expression profiles of patients with HCC revealed that 25 of the SLC-related genes were significantly dysregulated between high- and low-glycolytic tumors. Upregulation of *SLC2A1* (glucose transporter, fold change: 7.759), *SLC1A5* (neutral amino acid transporter, fold change: 2.132), and *SLC7A6* (cationic amino acid transporter, fold change: 1.666) and downregulation of *SLC16A1* (monocarboxylate transporter, MCT1, fold change: 0.57) and *SLC13A5* (citrate transporter, fold change: 0.498) in high-glycolytic HCCs (*n* = 8) compared to low-glycolytic HCCs (*n* = 8, Figure 1A) were observed. To confirm the results of the gene expression profiles, we performed TCGA analysis including HCCs from 360 patients. A significant negative correlation between *SLC2A1* and *SLC13A5* (*p* < 0.0001), as well as between *HIF1**α* and *SLC13A5* (*p* = 0.0158), and a positive correlation between *SLC16A1* and *SLC13A5* (*p* = 0.008) were revealed (Figure 1B–D and Appendix A). Furthermore, survival analysis using the database showed that *SLC2A1* (*p* < 0.0001) and *SLC13A5* (*p* = 0.0064), but not *SLC16A1* (*p* = 0.3773) expression was associated with OS (Figure 1E–G). *SLC13A5* and *SLC16A1* were highly expressed in low-glycolytic HCCs without HIF1α expression. Given the prognostic value for predicting OS, *SLC13A5* was more important than *SLC16A1*.

### 3.2. Citrate Treatment Affects HIF1α in HCC Cells with High NaCT Expression

The results from human data showed a significant negative correlation between NaCT and HIF1α expression, which raised the question of whether citrate affects HIF1α expression and glycolysis in HCCs. To further investigate the role of citrate, ^14^C-DG and ^14^C-citrate uptake were screened in human HCC cell lines. There was an inverse correlation between ^14^C-DG and ^14^C-citrate uptake among the HCC cell lines, in which HepG2 cells had the lowest ^14^C-DG but the highest ^14^C-citrate uptake (Figure 2A,B). When the hypoxic condition was induced, increased ^14^C-DG uptake and decreased ^14^C-citrate uptake were noted in HepG2 and Huh7 cells, respectively (Figure 2A,B). Consistent with the uptake assay, HepG2 showed the highest NaCT expression but the lowest hexokinase 2 (HK2) expression. However, the expression of SLCs was reversed in hypoxia (Figure 2C). When cells were transfected with si*HIF1**α*, reduced NaCT expression upon hypoxia was restored (Figure 2D). The results confirmed an inverse correlation between HIF1α and NaCT expression. Membranous GLUT1 is responsible for the constitutive uptake of glucose into cells [25]. HepG2 cells showed high cytoplasmic expression of GLUT1, which could not be differentiated from membranous GLUT1 on WB analysis. Therefore, HK2 was used for WB studies as a surrogate for glycolysis (Appendix A).

Next, to see the effects of extracellular citrate on HIF1α expression and tumor glycolysis, HCC cell lines were treated with citrate for 24 h under hypoxic conditions. Citrate treatment significantly reduced HIF1α and HK2 expression and increased NaCT expression in HepG2 cells. However, citrate treatment did not affect HIF1α and NaCT expression in Huh7 and SK-Hep1 cells with high glucose uptake and low NaCT expression (Figure 2E and Appendix A). Accordingly, hypoxia-induced increased ^14^C-DG uptake was significantly reduced (Figure 2F). These findings indicate that citrate treatment decreased hypoxia-induced glycolysis via reduced HIF1α expression in HCC cells with a high transport capacity for citrate.

### 3.3. Effects of Citrate Treatment on Glycolytic Pathway and HIF1α Regulation

To understand the effect of citrate treatment on cellular metabolism, we analyzed global gene expression profiles in HepG2 cells. The hierarchical clustering heat-map analysis was performed with dysregulated genes after treatment with 1 and 5 mM citrate under hypoxia compared to normoxia. Gene expression was dysregulated in a dose-dependent manner (Figure 3A). In fact, gene expression patterns of 5 mM citrate treatment under hypoxia resembled those of normoxia. In addition, we evaluated the biological functions of the 4334 dysregulated genes following 5 mM citrate treatment using Kyoto Encyclopedia of Genes and Genomes pathway analyses (Figure 3B). The dysregulated genes were mainly associated with the following categories: metabolic pathway, pathway in cancer, and phosphoinositide 3-kinase (PI3K)/protein kinase B (AKT) signaling pathway. Among the metabolic pathway, oxidative phosphorylation (*p* = 0.0), pantothenate and CoA biosynthesis (*p* = 0.008), glycerolipid (*p* = 0.01), and ether lipid (*p* = 0.04) metabolism pathways were upregulated, whereas hypoxia (*p* = 0.008), KRAS signaling (*p* = 0.04), and notch signaling (*p* = 0.012) pathways were downregulated after citrate treatment (Appendix A). Therefore, in cancer metabolism, mitochondria-related pathways are significantly regulated at the gene expression level by citrate treatment. In the metabolic pathway category, glycolytic metabolism-related genes (*HK1*, *PFKP*, *ALDOA*, *GAPDH*, and *PGK1*) were dose-dependently downregulated. In contrast, the genes related to the TCA cycle (*FH*, *SDHD*, *IDH1*, and *ACO1*) and fatty acid synthesis (*FADS1*, *FADS2*, *ELOVL2*, and *MCAT*) were upregulated (Figure 3C).

Finally, we investigated the underlying mechanisms on how citrate treatment affects HIF1α expression. Our analysis showed significant changes of expression in HIF1α stability-related gene (*PTEN*, *AKT*, *MDM2*, and *SIRT1*) upon citrate treatment. The upregulation of phosphatase and tensin homolog and MDM2, and downregulation of phosphorylated(p) AKT, pMDM2, and Sirtuin1 (SIRT1) were also identified in protein levels (Figure 3D). MDM2 is known to increase proteasomal degradation of HIF1α through PTEN/PI3K/AKT signaling [26]. Additionally, we found that MDM2-silenced HepG2 cells failed to degrade HIF1α (Figure 3E). Furthermore, HIF1α expression was partially reversed by MG-132, a proteasome inhibitor, suggesting the involvement of the ubiquitin–proteasome pathway (Figure 3F). Taken together, the results suggest that citrate treatment affected HIF1α expression at the protein stability level in the proteasomal pathway.

### 3.4. Anti-Tumor Effects of Citrate in Human HCC Cell Lines and Mouse Xenograft Models

We evaluated cell viability in hypoxia based on the impact of HIF1α degradation by citrate treatment. Cell viability was significantly reduced after 5 mM citrate treatment in HepG2 cells, but not in Huh7 and SK-Hep1 cells, with lower citrate transport capacity (Figure 4A and Appendix A). In addition, intracellular ROS levels and apoptosis were increased (Figure 4B,C and Appendix A), but intracellular ATP levels were significantly reduced (Figure 4D).

Next, we evaluated the anti-cancer effects of citrate in vivo. Mouse xenograft using HepG2 cells were allowed to drink water supplemented with the vehicle or citrate at 4 g/kg/day for 28 days (Figure 4E). Citrate treatment significantly reduced tumor volumes, and the growth rate was lower than that observed with vehicle treatment (−4.8%, Figure 4E). In addition, mouse xenografts in the citrate-treated group showed significantly smaller hypoxic regions than the tumors in the vehicle-treated group, consistent with the HIF1α degradation upon citrate treatment in the in vitro cell line experiments (Figure 4F,G and Appendix A). In addition, as in the in vitro experiment, p-AKT was downregulated, whereas PTEN and MDM2 was upregulated after citrate treatment (Appendix A). Moreover, the reduction of HIF1α-related protein (GLUT1), proliferation index (Ki67) and angiogenesis (CD31), and the increased level of cleaved caspase-3 (CC3) were observed in citrate-treated animal model (Appendix A). To summarize, citrate treatment showed anti-tumor effects in the in vitro cell lines and in vivo mouse xenografts.

## 4. Discussion

Increased glycolysis measured using ^18^F-FDG PET/CT is associated with hypoxia, which leads to resistance to chemotherapy and radiotherapy, as well as poor clinical outcomes in patients with HCC [27,28]. Thus, inhibiting HIF1α expression can ameliorate dysregulated glucose metabolism, which can effectively prevent tumor progression in HCCs. In this study, gene expression profiles in human HCCs showed an opposing pattern of SLC expression in tumor regions with or without increased ^18^F-FDG uptake. Tumor regions without increased ^18^F-FDG uptake showed increased *SLC13A5* and *SLC16A1* expression, whereas tumors with increased ^18^F-FDG uptake showed high *SLC2A1* and *HIF1**α*. Although MCT1 expression is already well-known, our study discovered high *SLC13A5* expression in HCCs of low glycolysis.

Metabolic phenotypes in tumors are related to the differing utilization of nutrients and aberrant expression of the nutrient transporters [8,29,30]. Although it might be important for developing therapeutic strategies, little is known about the metabolic characteristics and clinical significance of cancers using different nutrients. In human HCCs, it is already apparent that increased acetate metabolism through MCT1 expression plays an important role as an alternative carbon source of glucose for supporting tumor growth [14,31]. Compared to glycolytic tumors, HCCs with acetate utilization were less malignant and associated with better patient prognosis [32,33]. Previously, there was no information on whether acetate is the only carbon source in low-glycolytic HCCs. In this study, we found citrate to be another carbon source. *SLC13A5*/NaCT, a plasma membrane citrate transporter, has recently been identified as a metabolic target in human liver cancer cell lines [34]. However, the character of HCCs with increased *SLC13A5*/NaCT expression was unknown. Our result showed that increased citrate uptake was associated with decreased glucose uptake in in vitro cancer cell lines, mouse xenograft models, and human HCCs. More importantly, we found a significant negative correlation between NaCT and HIF1α expression in human HCC, suggesting that citrate can be potential therapeutic strategy preventing metabolic adaptation to hypoxic tumor. However, the mechanism of HIF1α-mediated regulation of *SLC13A5*/NaCT expression has not been addressed in this study.

In normoxia, citrate reduces glucose flux into cancer cells [21]. Here, we investigated the impact of extracellular citrate treatment on cancer metabolism under hypoxia. In HCC cells with NaCT expression, citrate treatment induced increased NaCT expression and intracellular citrate uptake, which downregulated the genes related to glycolysis despite hypoxia.

Next, we investigated how citrate induced the reduction of HIF1α under hypoxia. HIF1α levels can be controlled at the protein level by the ubiquitin–proteasome pathway and translational modifications, and at the mRNA level [35]. In this study, we found significantly increased MDM2 in the PTEN–PI3K–AKT–MDM2 pathway upon citrate treatment. MDM2, a ubiquitin ligase, binds to HIF1α for degradation by the ubiquitin–proteasome pathway under hypoxia [35]. In addition, phospho-MDM2 (pMDM2) translocation to the nucleus is mediated by the upregulation of phospho-AKT (pAKT) [26,36]. We assume that citrate treatment reduces pAKT activity (Figure 3D), resulting in the inhibition of pMDM2 translocation from the cytoplasm to the nucleus. An increased amount of unphosphorylated MDM2 in the cytoplasm may be involved in the proteasomal degradation of HIF1α. Gene silencing of MDM2 and treatment with proteasome inhibitor failed to degrade HIF1α upon citrate treatment, confirming the role of citrate in HIF1α degradation through 26S proteasomes. In addition, MDM2 increases the ubiquitination and proteasomal degradation of SIRT1. Given the role of SIRT1 in stabilizing HIF1α by direct binding and deacetylation in hypoxia [37,38], the decreased level of SIRT1 might contribute to the increased HIF1α degradation by MDM2. Taken together, we found a new regulatory mechanism in which citrate treatment reduced HIF1α expression through the regulation of protein degradation via increased MDM2 and decreased SIRT1 expression.

A few case reports have shown anti-tumor effect of oral citrate administration in patients [39,40]. Thereafter, studies have investigated the anti-cancer properties of citrate treatment and the potential mechanisms related to it [22,23,24]. In lung and breast cancer animal models, citrate treatments inhibited tumor growth by suppressing the insulin-like growth factor type 1 receptor (IGF-1R)/AKT/PTEN/eukaryotic translation initiation factor 2A (eIF2A) pathway [22]. Wang et al. reported similar results by inhibiting phosphofructokinase 1 (PFK1), a glycolytic enzyme in a gastric cancer model [23]. In this study, we also found that citrate treatment reduced tumor cell growth in an in vitro cell line and resulted in smaller and fewer hypoxic tumors with a decreased proliferative index in in vivo animal models. In contrast to previous studies, we observed that increased intracellular citrate uptake caused HIF1α degradation through the PTEN–PI3K–AKT–MDM2 pathway, responsible for reduced glucose flux. Accordingly, reduced intracellular ATP and increased ROS levels and apoptosis contributed to reduced cellular proliferation. Future studies are required to determine the potential of citrate treatment to overcome chemotherapy resistance.

## 5. Conclusions

There is a contrasting pattern of GLUT1 and NaCT expression in HCCs, meaning that cancer cells with low glucose uptake have high citrate uptake and vice versa. In turn, citrate treatment suppresses HIF1α stability and glycolysis, which can be a therapeutic strategy to prevent further hypoxic tumor progression and overcome chemotherapy resistance in low-glycolytic HCCs with citrate dependence.

## Figures and Tables

**Figure 1 cancers-14-03355-f001:**
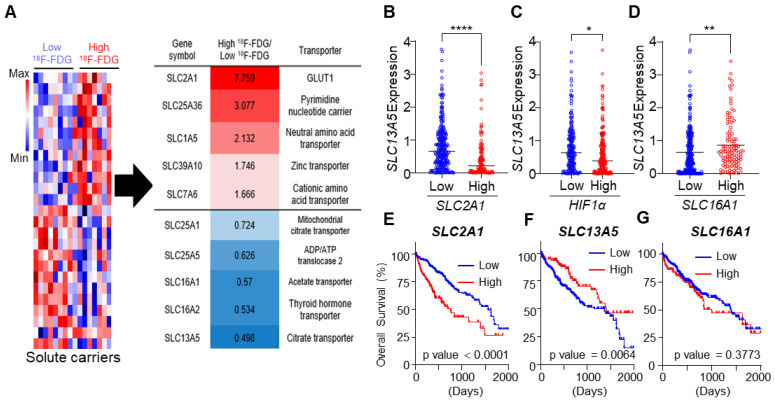
Reciprocal expression of solute carrier (SLC)-related gene in patients with hepatocellular carcinoma (HCC) with high-and low-glycolysis. (**A**) Heat-map of SLC-related genes of frozen tissues from patients with HCC with high ^18^F-fluorodeoxyglucose (FDG) (*n* = 8) or low ^18^F-FDG (*n* = 8). (**B**–**D**) The expression of *SLC13A5* with low and high *SLC2A1* (**B**)*,*
*HIF1**α* (**C**), and *SLC16A1* (**D**) expression groups (70-30 percentile) in the Cancer Genome Atlas (TCGA) dataset of HCC patients. (**E**–**G**) The median OS based on the TCGA dataset of HCC patients in low and high expression groups (70-30 percentile) in indicated genes. Data were represented as mean ± SEM and were compared using unpaired two-tailed *t*-tests and log-rank tests. * *p* < 0.05, ** *p* < 0.01, **** *p* < 0.0001.

**Figure 2 cancers-14-03355-f002:**
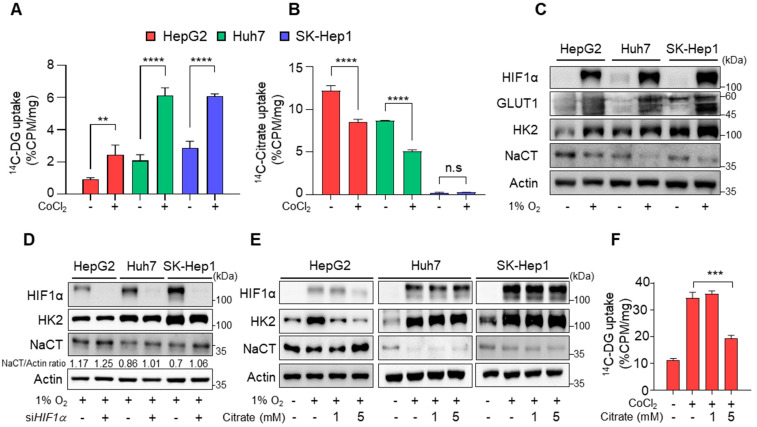
Reduction of glycolysis after citrate treatment. (**A**,**B**) In vitro ^14^C-DG uptake (**A**) and ^14^C-citrate (**B**) in HCC cells after incubation with or without 100 μM CoCl_2_. (**C**) The expression of HIF1α, GLUT1, HK2, NaCT, and β-actin in HCC cells after 24 h incubation under 21% or 1% O_2_. (**D**) HIF1α, HK2, NaCT, and β-actin expression levels in HCC cells after transfection with siRNA oligos against *HIF1**α* (si*HIF1**α*) or scrambled siRNA oligos for 24 h under 1% O_2_ conditions. (**E**) HIF1α, HK2, NaCT, and β-actin expression levels in HCC cells after treatment with the indicated condition for 24 h. (**F**) In vitro ^14^C-DG uptake in HepG2 cells after the indicated treatment. Data were represented as mean ± SEM and were compared using unpaired two-tailed *t*-tests and one-way ANOVA tests. ** *p* < 0.01, *** *p* < 0.001, **** *p* < 0.0001; n.s, not significant. Full Western Blot images can be found in Appendix A.

**Figure 3 cancers-14-03355-f003:**
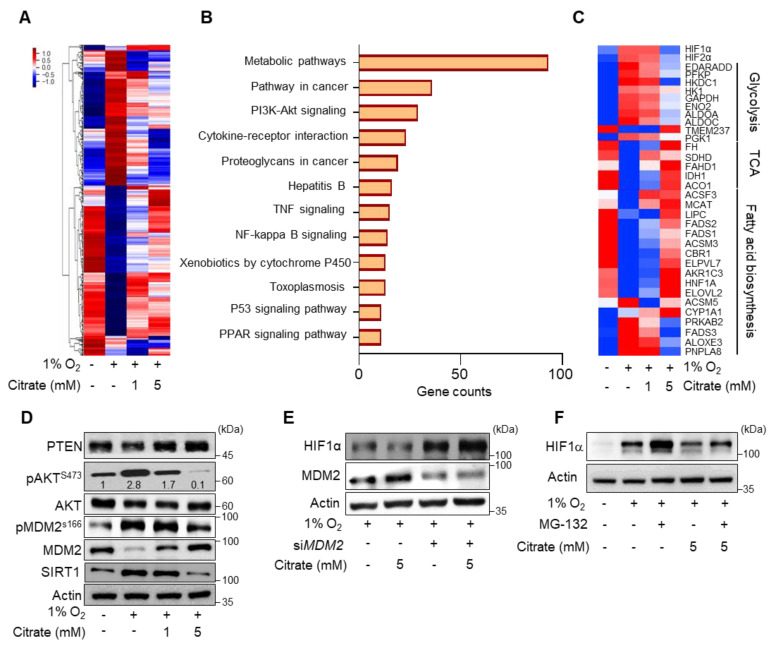
Effect of citrate treatment on HIF1α expression. (**A**) Heat-map representing color-coded expression levels of differentially expressed genes (*p* > 0.05, normalized data(log2) > 4, up- or downregulated > 1.5-fold) in HepG2 cells after treatment with or without citrate treatment under 1% O_2_ for 24 h. (**B**) Top 12 significantly enriched pathways of differentially expressed genes associated with citrate treatment under hypoxic conditions as analyzed by the Kyoto Encyclopedia of Genes and Genomes pathway analysis. (**C**) Heat-map of dysregulated genes related to glycolysis, TCA cycle, and fatty acid biosynthesis after treatment with the same as (**A**). (**D**) PTEN, phosphor (p) AKT (S473), AKT, MDM2, pMDM2 (S166), SIRT1, and β-actin expression levels in HepG2 cells after incubation with citrate (0, 1, or 5 mM) under 21% or 1% O_2_ for 24 h. (**E**) HIF1α, MDM2, and β-actin expression levels in HCC cells after transfection with siRNA oligos against *MDM2* (si*MDM2*) or scrambled siRNA oligos for 24 h incubation upon indicated condition. (**F**) HIF1α and β-actin expression levels in HepG2 cells after treatment with citrate and MG-132 under 21% or 1% O_2_ for 8 h. Full Western Blot images can be found in Appendix A.

**Figure 4 cancers-14-03355-f004:**
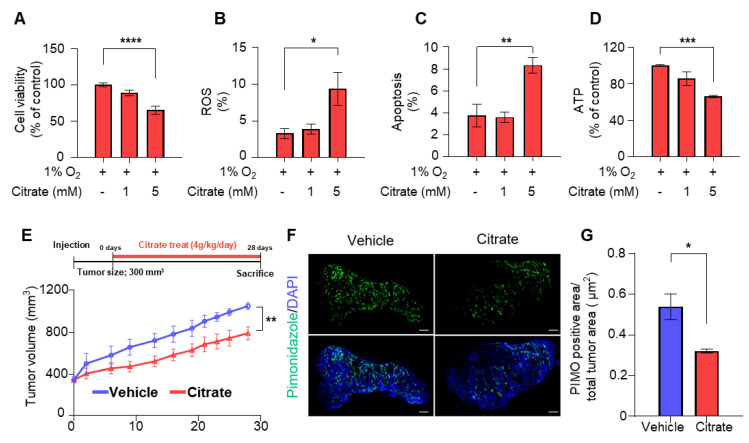
Anti-tumor effect of citrate in vitro and in vivo studies. (**A**) Cell viability in HepG2 cells following indicated treatment under 1% O_2_ for 24 h. (**B**) ROS production was measured by flow cytometry using MitoSOX staining in HepG2 cells following treatment with the same as (**A**). (**C**) Rates of apoptosis (%) counted by flow cytometry following staining with Annexin V-fluorescein isothiocyanate and PI in HepG2 cells after treatment with the same as (**A**). (**D**) ATP levels in HepG2 cells after treatment with the same as (**A**). (**E**) The average tumor volumes of mice receiving drinking water supplemented with vehicle (*n* = 5) or citrate (4 g/kg/day, *n* = 6) for 28 days. (**F**) Representative images of immunofluorescence staining with pimonidazole. (**G**) Quantification of the percentage of pimonidazole-positive area from vehicle (*n* = 4) or citrate-treated (*n* = 4) mice. Data represented as mean ± SEM and were compared using unpaired two-tailed *t*-tests and one-way ANOVA tests. * *p* < 0.05, ** *p* < 0.01, *** *p* < 0.001, **** *p* < 0.0001.

## Data Availability

All data generated or analyzed during this study are included in this article and its Appendix A. Further inquiries can be directed to the corresponding author.

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
