# Peer review of "Extracellular Citrate Treatment Induces HIF1α Degradation and Inhibits the Growth of Low-Glycolytic Hepatocellular Carcinoma under Hypoxia"

_cancers, 2022, doi:10.3390/cancers14143355_

Round 1

Reviewer 1 Report

In this article, Kim et al showed an anti-correlation between HIF1a expression and the citrate transporter, SLC13A5 in HCC. They further showed that under hypoxic condition, supplementation of citrate destabilized HIF1a and increased its degradation via increased MDM2 ubiquitin ligase activity. This leads to decreased glycolysis, increased apoptosis, increased intracellular ROS and reduced ATP level that eventually leads to decreased cell viability. The results and mechanisms are generally convincing. I recommend the following suggestions to further strengthen the manuscript.

·      Since the main theme of the article is about citrate, the authors should provide more information on how citrate supports metabolic function in tumors in the introduction.

·      The main title of the paper is about citrate as an alternative carbon source but there is no mention of how citrate is utilized in the cell. The title is misleading!

·      It would be more informative to show the negative correlation with correlation coefficient between high vs low glycolytic tumors for figure 1B-1D.

·      In figure 2D, do the authors mean HIF1a regulates NACT expression? If so, it is unclear how HIF1a regulates the level of SLC13A5/NACT level? This should be explained in detail. Also, the western blots should be quantified, as the differences are not significant in figure 1D.

·      Since the authors reported inverse correlation between NACT and HIF1a, do they expect to see decreased HIF1a after siSLC13A5 even in the presence of citrate under hypoxia? This shows that the effect is specific towards SLC13A5.

·      Figure 2E, the authors should comment on why Huh7 and SK-Hep1 have no effect on NACT after citrate treatment. They didn’t mention anything about the other 2 cell lines (section 3.2).

·      The authors investigated the level of HIF1a throughout the paper. How about HIF2a? Does it compensate for HIF1a? They should test the level of HIF2a. under these conditions.

·      The authors should annotate which pathways are upregulated and which pathways are downregulated with citrate treatment (figure 3B).

·      In figure 3D. do the authors suggest that citrate supplementation inhibits AKT activity? Also, AKT activity has been shown to positively regulate MDM2 protein level. The authors showed opposite trend between pAKT and MDM2. Is this phenotype specific to hypoxia? If so, this should be explained in detail and referenced.

·      The decrease in cell viability after citrate treatment (figure 4A) is too modest to have a significant effect. Also, the author reported an increase in ROS level after citrate treatment. They should speculate and discuss how citrate treatment induces ROS and decreases ATP levels.

·      In figure 4F, the authors reported that citrate treatment decreased the hypoxic regions. Do they mean with citrate treatment, all the hypoxic cells undergo apoptosis? If so, the tissue should be stained for CC3. This would further support their in vitro findings as well.

Minor comments:

·      The authors should explain the criteria for high vs low glycolytic tumor.

·      Figure S1 and S2 are not referenced in the results section.

Reviewer 2 Report

The authors show an indirect relationship between NaCT and HIF-1 expression in human HCC samples. Using HCC cells, hypoxia was demonstrated to decrease NaCT in a HIF-1 dependent manner. Citrate is shown to inhibit HIF-1 via proteasomal degradation. Further, the xenograft model was assessed to show that citrate inhibits tumor growth. Based on these evidence, the authors conclude that citrate inhibits glycolysis in HCC by degrading HIF1a. However, there are several issues with the data and interpretation.

1.     The authors show that the mRNA levels of NaCT and HIF-1 are inversely proportional. However, the authors show that citrate regulates HIF-1 expression via MDM2-mediated proteasomal degradation. What regulates the transcription of HIF-1 in human HCC?

2.     The reversal of NaCT by si-HIF1 shown in Figure 2D is not convincing. The authors should provide quantitation of the Westerns.

3.     Does knockdown of MDM2 restore NaCT?

4.     Does knockdown of MDM2 induce a glycolytic switch in HCC cells?

5.     Pimonidazole marks the hypoxic region and should not be interpreted as HIF-1 expression. Moreover, low pimonidazole staining is expected in smaller tumors. The authors should show HIF-1 expression by Western blot or immunostaining.

6.     Moreover, the authors should examine the expression of HIF-1 target genes in these xenografts (including glycolytic genes).

7.     Citrate fluxes through the TCA cycle. Is the citrate or TCA metabolite regulate MDM2 or HIF-1 expression. Whats the mechanism?

8.     The authors should measure the citrate levels in tumors.

9.     The expression of PTEN, pAKT and MDM2 should also be shown in the citrate treated HCC.

10.  Based on the data provided, it is unclear which comes first. Whether an induction in HIF-1 (via tumor hypoxia) decreases NaCT leading to a glycolytic switch or a decrease in NaCT occurs first to stabilize HIF-1 in HCC. A model of the proposed study may be helpful.

Reviewer 3 Report

Hypoxic tumors are more resistant to cell death stimuli and chemotherapy. Kim et. al., identified SLC13A5/NaCT as an important solute carrier (SLC) in low-glycolytic HCC. With extracellular citrate treatment the cell might not switch from oxidative phosphorylation to hypoxia-induced glycolysis . The authors also established an inverse correlation between HIF-1a and NACT expression. They also demonstrated how citrate treatment suppresses HIF1α stability and glycolysis and can be exploited as potential target for overcoming hypoxia tumor progression and chemotherapy resistance which can be very beneficial for the researchers working in cancer field. I have few minor comments:

1.      Authors are suggested to mention the catalog number of Lipofactamine used for transfection

2.      Authors are suggested to rephrase line 101, Pg 3, “For hypoxia experiments, a hypoxia incubator (1% O2) or 100 μM CoCl2 was used as necessary…”

3.      For in vitro 14C-deoxyglucose (DG) and 14C-citrate uptake assay, which low  glucose medium was used and what was the reaction buffer

4.      Was there any specific reason for including female mice only for in vivo studies and what was the count of them?

5.      Since HKII is normally associated with VDAC, did authors checked the effect of citrate treatment on the VDAC?

Round 2

Reviewer 1 Report

The authors have addressed all my comments and I recommend the manuscript for publication in Cancers.

Reviewer 2 Report

No further comments.